# Intelligent Process Automation and Business Continuity: Areas for Future Research

**José Brás** [1,2,*] , **Ruben Pereira** [1] **and Sérgio Moro** [1]

1 Department of Information Science and Technology, Instituto Universitário de Lisboa (ISCTE-IUL), ISTAR, 999022 Lisboa, Portugal
2 CGI Innovation Hub Lisbon, 999022 Lisboa, Portugal
* Correspondence: jose_manuel_bras@iscte-iul.pt

**Abstract:** Robotic process automation and intelligent process automation have gained a foothold in the automation of business processes, using blocks of software (bots). These agents interact with systems through interfaces, replacing human intervention with the aim of improving efficiency, reducing costs and mitigating risks by ensuring and enforcing compliance measures. However, there are aspects of the incorporation of this new reality within the business continuity lifecycle that are still unclear, and which need to be evaluated. This study provides a multivocal literature review of robotic process automation and intelligent process automation correlated with business continuity, to identify the level of awareness of these two emerging forms of automation within the business continuity management lifecycle. Based on the reviewed literature, the study develops a discussion of the main research areas for investigation, identifying what is attracting the attention of practitioners and researchers and which areas they highlight as promising for future research. Numerous sources from relevant backgrounds reveal an interest in these interrelated topics but there as yet is little or no information available on the direct connection between them.

**Keywords:** business continuity; governance; risk; compliance; robotic process automation; intelligent process automation; business process management

## 1. Introduction

Companies have been on a mission to digitize operations for years, but recent global events, notably the COVID-19 pandemic, have accelerated the process of digital transformation (DT) [1,2] to help organizations grow and cope with instability and disruptions to businesses. Trends such as hyperautomation [3] and hyperconnectivity [4,5] leverage an ever-growing hyperconnected society [6,7] and companies are adopting automated solutions to execute and modernize their business processes (BP) [8] and help ensure business continuity (BC). With automation becoming a new norm for organizations to support their growth and cost optimization strategies, more and more emerging technologies (ET) associated with automation are being adopted, such as robotic process automation (RPA)/intelligent process automation (IPA) [9], intelligent automation (IA) [10], artificial intelligence (AI), and AI-based decision-making tools [11], among others.

RPA/IPA have gained momentum [12], offering solutions to achieve efficiency gains [13] or mitigate organizational problems [14,15]. Figure 1 illustrates the growing interest in the topic, expressed in terms of search results for the keywords from 2010 to 2022 using the Google search engine, thus showing the attention that they have received since 2010.

However, the advent of automation, which potentiates efficiency gains and also resolves problems that result from a lack of human resources, can also create new challenges in terms of dealing with new risks that are still not fully understood [16,17]. Therefore, its impacts on BC must be properly evaluated and require further investigation [15,18,19]. Our research aims to answer the following question: What are the most important areas to investigate in the future with regard to BC and RPA/IPA?

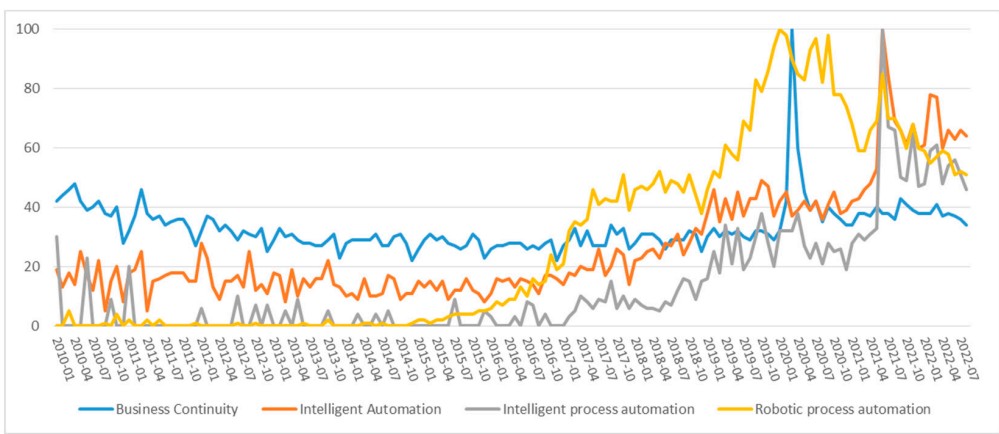

**Figure 1.** Google trends for BC, RPA, IPA and IA, 2010–2022 (adapted from [20–23]).

The possible need to adopt new procedures for the introduction of new technologies in such a rapid and sometimes disruptive way needs to be properly evaluated in order to adapt and prepare BC for the incorporation of ET such as IPA and RPA. This involves understanding how to promote and develop more flexible recovery strategies appropriate for the new realities, as business processes are now in an almost constant state of change [24,25]. Therefore, it is important to provide companies with insights into how BC professionals can handle business disruptions affecting BP that rely solely on RPA and how to take advantage of this technology to make BC more predictive and less responsive, avoiding disasters by using AI-powered software that can perform these BC-related tasks [26].

This research aims to determine the most important areas to investigate in the future with regard to BC and RPA/IPA. Drawn upon a large body of knowledge, we provide an understanding of the impact on BC resulting from their introduction in organizations, as these two areas together (RPA/IPA and BC) open up a multitude of unknowns that need to be investigated [27–32].

## 2. Background

### 2.1. Business Continuity Management

In comparison to the majority of other business management disciplines, business continuity management (BCM) [33,34] is relatively new, as it first appeared in the 1960s as information technology (IT) "disaster recovery" to safeguard company investments in technology; it then gradually evolved, grounded in emerging legislation and standards until 2001 [35]. Figure 2 illustrates the evolution and the major milestones in this process.

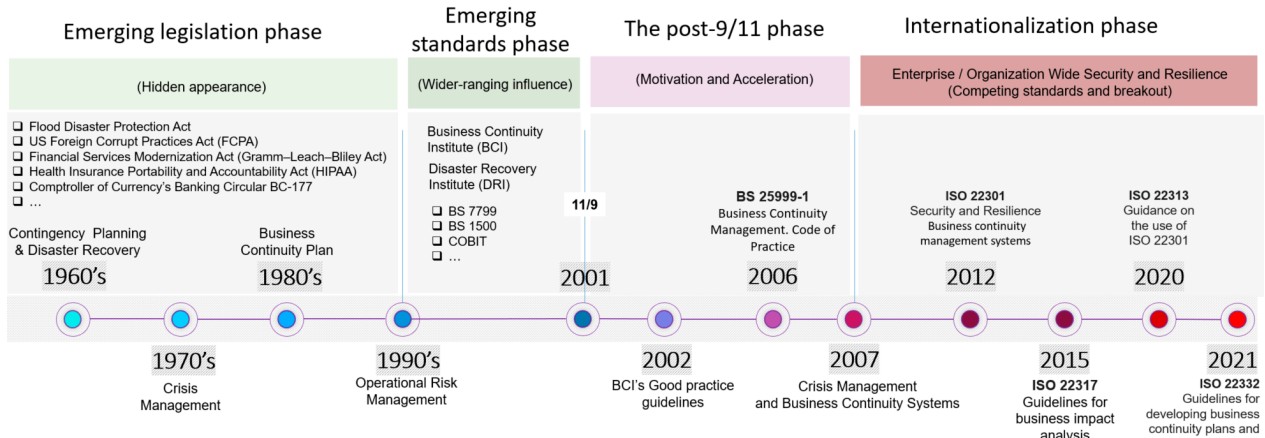

**Figure 2.** BCM evolution phases (adapted from [27,36]).

Business continuity has become more relevant over time due to disruptive events that have been affecting our lives and businesses: 9/11 in the USA [37], the Islamic state (ISIS) terrorist attacks throughout the world [38], climate changes that affect the planet [39,40], and the recent COVID-19 pandemic [41], which have all shaped the path of BCM (Figure 2). Due to the increasing risk, security, resilience and business continuity [34,42,43] are topics under the radar of corporate consulting services [18,44–48] and industry regulators [33,49]. However, as a result of a larger and constantly hypercompetitive business landscape [2,50], the forces of nature and all other threats present challenges that make it quite difficult for organizations to find objective and consistently effective ways to become resilient and pursue BCM. The level of resilience of an organization or its state of preparedness in terms of facing disasters is crucial to business continuity and IT disaster recovery, as this can mean the difference between success or failure of a company.

A business continuity plan (BCP) and disaster recovery plan (DRP) are not defined in the same way but have similar purposes: both aim to keep an organization operational without disruptions. The BCP is more dedicated to planning the recovery of processes and business functions [19], covering emergency response, business operations continuity, IT disaster recovery and crisis management. The IT DRP is a subset of business continuity, as it is the technical component of the BCP that addresses the recovery of core systems and their data, and enables information and communication technologies (ICT) to continue to operate and support the business.

Business continuity mainly establishes the strategies, procedures and critical actions required to successfully respond to a crisis situation [34]. In addition, it evaluates how well an organization can respond to unexpected disasters, disruptions or sudden changes to its business environment [28,29]. As crises may result from a natural disaster, a catastrophe or just a simple accident that can interrupt services, resulting in the partial or total loss of business [28], a BCP must address all possible situations in order to mitigate or assume undertaken risks.

The British Standards Institution (BCI) defines business continuity as the "capability of the organization to continue the delivery of products or services at acceptable predefined levels following a disruptive event" [29]. It also defines BCM as a "holistic management process that identifies potential threats to an organization and the impacts to business operations those threats, if realized, might cause, and which provides a framework for building organizational resilience with the capability for an effective response that safeguards the interests of its key stakeholders, reputation, brand and value-creating activities" [34]. Furthermore, ISO 22301 [51] specifies that the purpose of a BCP is to provide a documented framework and processes to enable an organization to synthesize all of its business processes within its recovery time objective after a disruptive incident [36].

It is fundamental to define the entire critical process and all the elements needed to perform these tasks in order to ensure business continuity and organizational resilience [30]. One of the main challenges involves proactively elaborating, developing and implementing BCP and DRP and establishing the required knowledge of all the key resources, key activities and key elements [52]. For example, after a disruptive event, an organization needs to establish redesign and re-engineering processes in order to adapt their business to new realities. It is important to note that in a crisis, an organization should have ways, means and tools that enable its operations to continue immediately [34].

To sum up, BCM is an essential tool that aims to ensure that the company is prepared for immediate recovery of its critical activities and their support systems and applications in the event of a disruption.

### 2.2. Robotic Process Automation and Intelligent Process Automation

These two terms appear to mean the same thing. However, while RPA focuses on automating repetitive tasks and processes based solely on rules, intelligent automation by its very nature incorporates a vast array of ET, such as AI, machine learning (ML),

natural language processing (NLP), structured data interaction, and intelligent document processing [53].

As a relatively recent technology, RPA is preconfigured software that is used to automate a combination of processes, tasks, activities or services, with graphical user interfaces that are choreographed to interact with almost any type of system as a human user would [53,54]. Both scientific research and the media highlight the potential of RPA for increasing the efficiency of processes [55–58]. Since the development of RPA solutions requires low levels of programming experience, with low implementation costs and a very fast return on investments, these solutions are suitable for an extensive wide range of processes and result in very high efficiency gains. Hence, the technology has attracted interest in the business world, with several examples of successful implementation [59].

The initial approach to this technology (intelligent process automation) involved replacing the routine or strictly transactional processes—previously performed by humans and now replaced by RPAs—with solutions capable of performing more complex tasks. As this technology evolved, capabilities related to ML and cognitive computing were added with increasingly sophisticated rule mechanisms, as a result of which it started to be able to perform more complex tasks, including evaluation, reasoning, decision-making and compliance with probabilistic and/or deterministic process requirements in dynamic contexts. Intelligent process automation is thus the evolution of simpler repetitive tasks, in which new capabilities are added with more sophisticated and complex procedures [53,60].

### 2.3. Interlinking Intelligent Process Automation with Business Continuity

In order to fully benefit from automation and address risks, failures or potential threats, organizations need to take a holistic approach to managing change, including alignment between business and IT, BC, and new controls designed to tackle the specific risks emerging from RPA/IPA. Figure 3 summarizes the key activities involving both domains, expressed in actions that illustrate key inter-domain touchpoints for IPA/RPA and BC. It shows the relationship between the two sets of requirements, one for implementation practice and management methodology for intelligent software-based process automation found in IEEE 2755.2-2020, and the other, ISO 22301-2019, pertaining to how the business can implement, maintain and improve a management system to protect from and reduce the likelihood of incidents, and prepare, respond to, and recover from outages when they arise. As this is ongoing research, it is important to find out what should be addressed in order to understand the impacts of the relationship between intelligent process automation and business continuity. Both must be analyzed so that users can develop intelligent software-based process automation or adopt BC procedures that meet the needs of both frameworks.

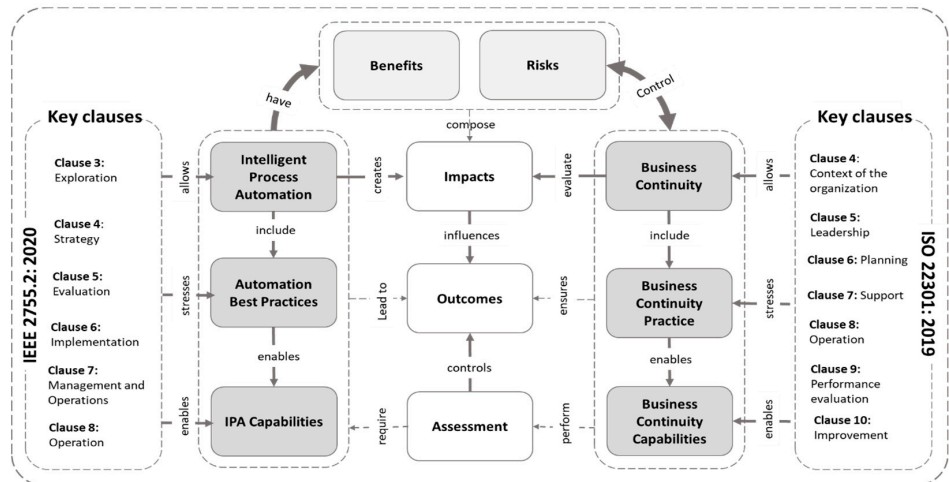

**Figure 3.** Inter-domain touchpoints for IPA/RPA and BC.

COVID-19 proved that the early adopters who invested in RPA already experienced key returns in 2020, due to the fundamental capabilities that RPA provided for organizations, since RPA can decrease the amount of work for humans, help manage the organization and ensure the continuity of the business [15,61,62]. Although this transformation gained strength through the gains obtained, it created a more complex ecosystem in which a great number of BPs have been updated to incorporate ET associated with RPA, requiring that all risks and benefits need to be re-evaluated to be controlled in terms of BC [63].

Despite academic interest, the topic has received more attention from industry, with several reports being published by various business and senior consulting companies such as Deloitte [61,64], Ernst & Young [65–67], KPMG [68–70] and PwC [63,71,72]. An assessment of the adequacy of BC processes and capabilities/practices to mitigate and support the risks raised by RPA activities is therefore required.

Given the increasing adoption of RPA, the predominant interest of professionals and the lack of systematization and understanding of the main areas of research in RPA and BC, a multivocal literature review (MLR) was conducted to identify the main areas for further investigation.

## 3. Research Methodology

### 3.1. Methods

The review protocol specifies the research question being addressed and the methods that are used to perform the review. In order to find the maximum number of studies related to the research question, a search strategy was used to detect as much of the relevant literature as possible using multiple keywords and datasets.

The research was carried out by using search strings to search for information on the main topic, "business continuity", associating it with other interrelated keywords (intelligent automation, intelligent process automation, automation, and RPA). With regard to academic data sources, the publications domain was identified by searching several electronic bibliographic databases, listed below, to build the datasets. The papers were collected on the basis of their title, keywords, abstract, submission for review and publication in academic journals. Google Search (www.google.com (accessed 22 August 2021)) was chosen to search for grey literature.

#### 3.1.1. Data Source and Searches

The PRISMA (preferred reporting items for systematic reviews and meta-analyses) guidelines were followed in the conduct and reporting of this systematic review.

The articles were collected between March and August 2022; and restrictions were applied regarding language (only English) and dates between 2017 and 2022. The following keywords were applied to the search: "business continuity" AND ("robotic process automation" OR "intelligent automation" OR "RPA" OR "IPA" OR "automation" OR "intelligent process automation"). Bibliographies from relevant publications were checked to identify relevant articles.

We searched the following databases for eligible studies:

1. IEEE Xplore Digital Library (https://ieeexplore.ieee.org/Xplore/home.jsp (accessed on 14 June 2021));
2. ACM (https://dl.acm.org (accessed on 1 July 2021)).
3. SpringerLink (https://link.springer.com (accessed on 5 September 2021))
4. Scopus (https://www.scopus.com/home.uri (accessed on 11 November 2021))
5. Web of Science (https://www.webofscience.com/wos/woscc/basic-search (accessed on 2 December 2021))
6. EBSCO (http://search.ebscohost.com/ (accessed on 27 December 2021))
7. Google Search (https://www.google.com/ (accessed on 22 August 2021))

We considered Google Search a limitation in terms of the replicability of the searches performed at a given time but, according to some authors [73], website search methods may differ and it is more important to have a considered rationale for the process, taking

the goals and objectives of each review into account, rather than specifying a single method. The planning and execution of the research, as well as the screening of results and the structure of its management, must be properly organized for this type of approach [73]. They recommend performing a grey literature search using at least one traditional search engine (e.g., Google, Yahoo or Bing) with the first 12 pages (instead of the first 5 pages) and an accurate search of academic databases that are more closely aligned with the topic under analysis, in order to ensure that all the relevant literature is considered and that the conclusions are more comprehensive [74,75].

### 3.1.2. Eligibility Criteria

For the qualitative analysis, we included articles related to main keywords (process automation or business continuity), present in the title, abstract, key contents or subject relevance. They were found in journals, conference papers, blogs or grey literature (limited to the first 12 pages of Google Search).

### 3.1.3. Study Selection

In the initial search stage (first filtration, shown in Figure 4), the filtering criteria—inclusion and exclusion criteria filters (all fields; all documents and full text, abstract, reviewed publications in journals, academic journals and grey literature)—were used together with the search string. This step is illustrated in Table 1, as part of the full MLR protocol to find the final sample for the elaboration of the article, which produces a list of the articles found, together with the filters used. All publications that met the inclusion criteria were selected and analyzed.

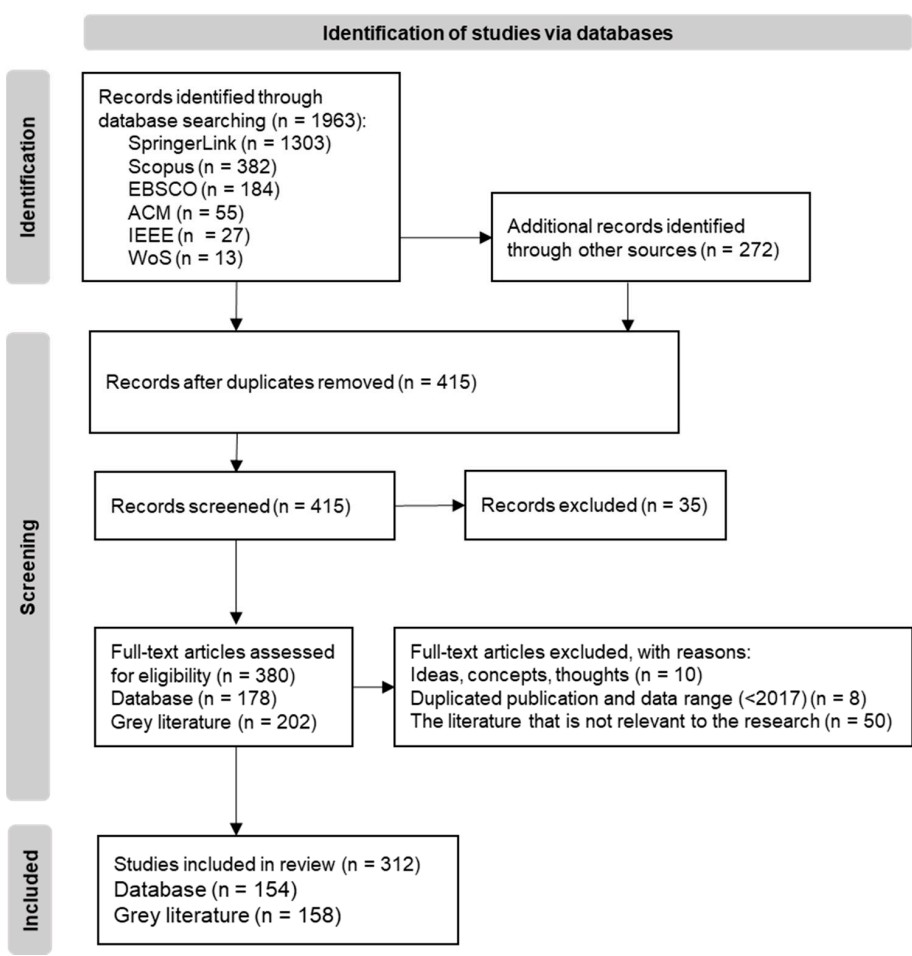

**Figure 4.** PRISMA Flowchart (adapted from [76]).

**Table 1.** Inclusion and exclusion criteria used.

| Inclusion Criteria | Exclusion Criteria |
| --- | --- |
| Related to main keywords | Not related to RPA, IA or IPA or business continuity |
| Process automation or business continuity | Paper not in English |
| Title, abstract, key contents or subject relevance | Documents with publication date earlier than 2017 |
| Journals, conference papers, blogs or grey literature | Vendor tool advertisements |
| Documents in English | Papers by unidentified authors |
| Limit results to first 12 pages of Google Search | No publication date |

In the case of the Google search engine, we consider it to be a valid source of grey literature, governmental and institutional reports. Although Google Search has its limitations and should not be used as the only source for systematic reviews, it was used here as it can be suitable for the purposes of qualitative systematic reviews. For the initial results, only the first twelve pages of the results were counted, which were then used for review and selection [75].

The study has the following research question: What are the most important areas to investigate in the future regarding BC and RPA/IPA?

The overview of the review process can be found in Figure 4, which provides a visual representation of the study selection process that was applied. This diagram represents the different selection steps used in the systematization of the selection process.

An inclusion and exclusion criteria was adopted in order to identify the relevant literature for this study. The screening criteria for including or excluding articles for this research are summarized and illustrated in Table 1.

In order to ensure, whenever possible, the inclusion of all relevant sources, backward and forward snowballing was applied to the set of articles already in the set, as recommended by the systematic review guidelines [77]. Snowballing, in this context, refers to using an article's reference list (backward snowballing) or article citations to identify additional articles (forward snowballing) [77].

A software package (Mendeley) was used to facilitate the task of searching and collecting the literature. This ensures that unique results are obtained, as the software detects and eliminates duplicate entries, thus solving the problem of consistency in the returned and collected results and also organizing it into different sets according to query strings and the academic or grey literature categories. Finally, it facilitates the work of retrieving the results of the distinct ID sets (academic and grey literature) that are easily merged in the study process.

## 4. Multivocal Literature Review

The multivocal literature review (MLR) [78] is similar to the systematic literature review (SLR) [79,80] and aims to incorporate the so-called "grey literature" in order to supplement the published (formal) literature. MLRs are SLRs which include both scholarly writing (also known as academic writing or formal writing) and the (informal) grey literature (GL) which is not considered in the SLR. GL is considered to be a multisource of information, which may exist in the form of blogs, videos, webpages and white papers that are produced outside academic forums and are not subject to any quality control mechanism (e.g., the peer review process) prior to publication.

By including information that normally would not be taken into account due to its "grey" nature [78], MLRs are important for the completeness of the research. An MLR in a given subject field is essentially a combination of the sources that would be studied in an SLR and a GLR in the same field. Thus, an MLR is, in principle, expected to provide a more complete picture of the evidence in a given field. Figure 5 represents the relationship between SLR, GLR and MLR.

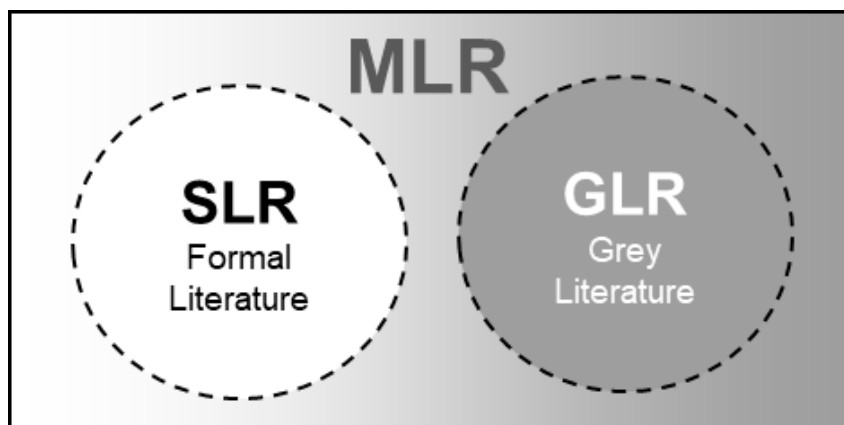

**Figure 5.** The relationship between SLR, GLR and MLR [78].

The objective is to explore the results of the MLR to provide a state-of-the-art overview of current work on this topic and to identify the most important research areas for BC using IPA, in order to:

1. Identify the most important areas for future research on incorporating RPA/IPA into BC;
2. Outline the definition of objectives for the research;
3. Prepare future surveys and interviews to evaluate the results found, and compare the objectives and the actual observed results from the use of existing projects;
4. Communicate the challenges and opportunities found, the results achieved, and their usefulness for other researchers and professionals (BC practitioners and the RPA/IPA community).

Table 2 distinguishes between "white literature" and "grey literature", listing the appropriate choice of publications in each case. "Black" or other types of literature subject to exclusion are also classified, to clarify the choices made during the assessment.

**Table 2.** Spectrum of "white", "grey" and excluded literature (adapted from [78]).

| "White" Literature | "Grey" Literature | "Black" or Other Types of Literature (Excluded) |
|---|---|---|
| Papers published in journals<br>Conference proceedings<br>Books | Preprints<br>e-Prints<br>Lectures<br>Datasets<br>Government documents<br>Standards<br>White papers<br>Technical reports<br>Blogs<br>Audio-video media | Ideas<br>Concepts<br>Thoughts |

The MLR workflow is summarized in Figure 6 and has three phases. The initial phase of the research ("planning the MLR") comprises two steps:

- Determining the need for an MLR for the given topic;
- Defining the MLR goal and setting up the research questions.

Once the MLR is planned, we proceed to the next phase of the research, namely "conducting the MLR". This phase is divided into five stages:

- Search process and selection: identification of primary studies to address the research question, application of standard comprehensive search techniques by means of defined search strings, and definition of the selection criteria for performing the selection process;

- Study quality: assessment of sources to determine the extent to which a source is valid and free from bias;
- Design of data extraction: creation of forms to gather all the information needed to address the review question and the study quality criteria;
- Data extraction: extraction of the data items needed to answer the research questions;
- Data synthesis: synthesis of data in such a way that the question(s) can be answered.

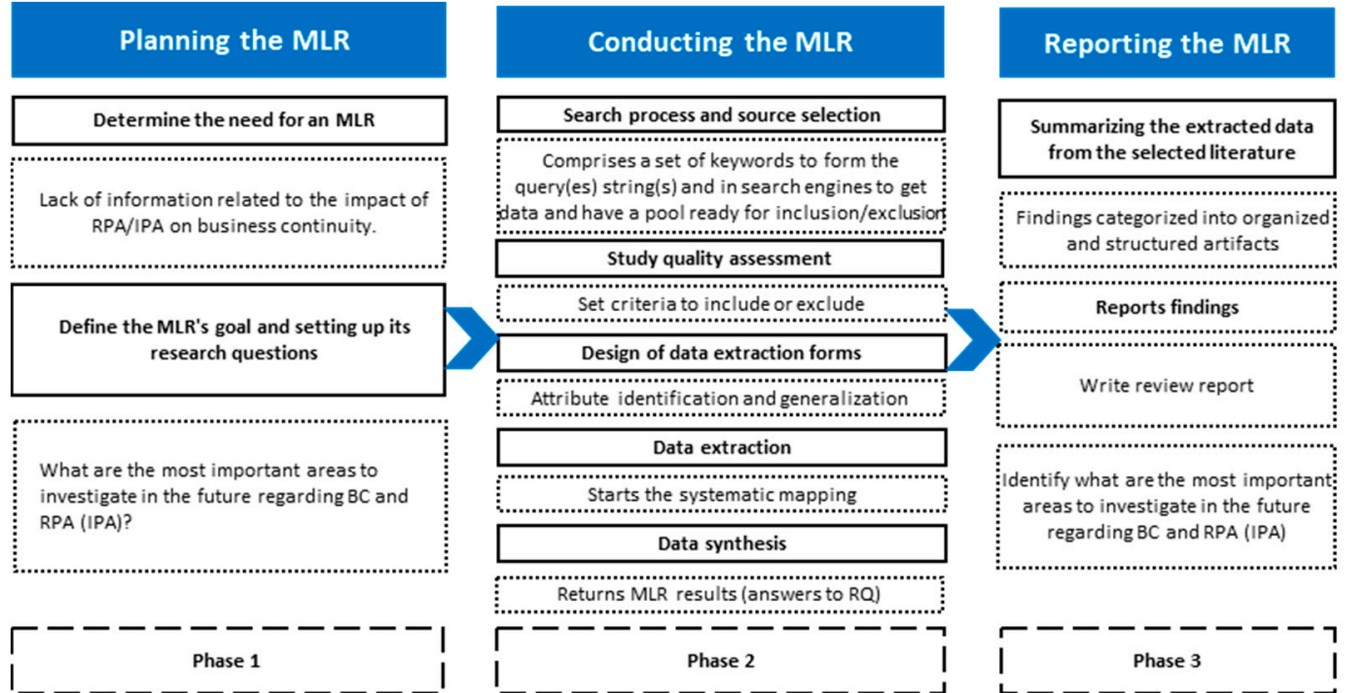

**Figure 6.** Multivocal literature review (MLR) phases and steps adopted in this research (adapted from [78]).

"Reporting the MLR" is the final phase and is very similar to the SLR guidelines provided by Kitchenham and Charters [81] for planning the MLR, specifying dissemination mechanisms, formatting the main report and evaluating the report.

## 5. Planning the MLR

*Motivation*

The COVID-19 pandemic had an impact on organizations, forcing huge changes in several areas which had to be made very quickly. Organizations needed to accelerate the use of digital technology and replace various business processes previously performed in what may be called a "traditional" way with alternative complementary ways of using technology [82]. The most striking examples were related to work roles that were downsized or replaced with technology as a technique to mitigate risks related to infection, while maintaining productivity. Online channels have become the salvation, both for consumers looking for products and also for companies looking for alternative ways to place their products and services on the market. There is a need to implement digital transformation solutions derived from the current state, which is in continuous progress.

However, organizations can also leverage technology to provide business process and service innovations and ensure business continuity [15].

Hence, RPA/IPA have recently become key drivers of digital transformation, supporting process automation more efficiently by replacing repetitive tasks performed by humans and ultimately helping business continuity. Within the domain of business process management, process automation aims to improve the company's workflows, reducing

costs, time and waste, as well as increasing productivity, and reducing errors in processes with technological support, through the application of robotic software to execute tasks.

Another aspect of the motivation for further research in this domain is the fact that RPA/IPA is being widely implemented in many types of organizations, transforming the complexity of the IT environment and business ecosystem even more. The response needed to face new threats and the need to deal with continuous challenges associated with BC is the driving force behind the eventual need to adopt new procedures in order to prepare BC to incorporate RPA and build good flexible recovery strategies [25]. However, management and practitioners need relevant information to support them in incorporating RPA/IPA into BC.

RPA has been attracting a lot of attention from the corporate world. However, although it is a popular topic there, academic research lacks in-depth theoretical analysis of RPA [83]. As a niche area of IT, literature on RPA is rather sparse in terms of its impact on organizations and consequently on business continuity. Further investigation should involve a comprehensive assessment of this technology, identifying guidelines for effective adoption and management and revealing additional factors, as well as the impacts that influence adoption of RPA technology [84].

Since RPA/IPA is being explored more within industry than academia, resulting in important inputs from the former [52,85–87], the MLR expands the origins of the sources to identify and map the most important areas for future investigation related to these two topics.

## 6. Conducting the MLR

This section describes how the review was conducted, which is the second phase of the process. In this stage, the research is carried out by searching for information in selected databases using the pre-defined queries and analyzing the extracted data.

## 7. Reporting the MLR

This section presents the organization of the research findings on the most important areas to investigate in the future regarding BC and RPA/IPA.

Our objective is to explore what the (scientific and grey) literature offers in terms of clues to future areas to be investigated. The results were obtained by analyzing and compiling the outcomes found in the results and future work sections of the literature that was analyzed, which led to three meta-themes or clusters: governance, risk management and compliance (GRC); people, processes and technology (PPT); and business continuity (BC). These clusters point to the need for further investigation in order to develop an even more detailed structural research approach to RPA/IPA in combination with business continuity.

Topics related to GRC are summarized in Table 3, in which the "count" column indicates the number of articles found during the investigation that are related to a given topic relevant to each meta theme. Table 4 shows the key findings related to PPT, and Table 5 shows topics related to BC, revealing a direct concern in the literature with how to provide resilient solutions to enable businesses to continue to operate in the event of disruption.

Governance/risk and compliance (Table 3)—commonly known as GRC—is a set of processes and procedures that aim to define a set of rules to assess the activity of organizations (audit procedures, policies and strategic management) to help achieve secure efficiency gains for business objectives, dealing with uncertainty and compliance [16,88,89].

Although it is not a new concept, GRC gained importance as the different types of risks became more numerous, diverse, complex and damaging, thus making it necessary to assess the maturity status of organizations [4,90–93].

The literature that was evaluated also indicated that there is a growing interest in looking for monitoring solutions and a need for new controls to regulate activities in view of the changes imposed by the introduction of RPA and to ensure compliance with the regulatory standards [17,64,94].

Table 4 represents the people/processes/technology grouping (PPT) [95] and its sub-themes which, as autonomous components, are fundamental to organizational transformation and management. In order to achieve organizational efficiency, organizations need to balance these three components and maintain a good relationship between them.

This grouping refers to the methodology in which the balance of people/processes/technology produces outcomes: people perform a specific type of work (using certain capabilities) for an organization using processes (and often technology) to streamline and improve processes [1,3]. This framework helps to achieve balance within an organization and is most often used when deciding whether to purchase or implement new technologies [68]. As these processes now often include different types of technology for diverse solutions, they can represent an increased risk and need to be evaluated according to the appropriate standards and guidelines [42,63,96].

**Table 3.** Group of key findings related to governance, risk and compliance.

| | Key Areas/Subjects for Investigation | Count * |
|---|---|---|
| Governance | Types (e-governance/corporate/non-corporate/etc.) | 26 |
| | Audit | 18 |
| | Policies | 13 |
| | Monitoring tools (KPI/Risks) | 12 |
| | Strategic management | 11 |
| | Productivity | 6 |
| | Efficiency gains | 4 |
| Risk | Types (avoidance/cyber/digital/financial/environmental/monitoring/ organizational/operational/profiling/health and safety/intellectual property protection) | 71 |
| | Risk assessment/management | 33 |
| | Cybersecurity | 29 |
| | Security (assessment/data/mindset/management) | 14 |
| | Privacy and security | 6 |
| | Enterprise risk management | 5 |
| | Security orchestration research and practice | 2 |
| Compliance | Monitoring | 25 |
| | Regulatory | 25 |
| | Controls | 17 |
| | Regulatory framework | 8 |

* Number of articles found related to GRC and their subtopics.

Technology is nothing without the right people using the right process to help them and the right guidelines to back them up. Thus, technology should always be the final consideration once a problem is clearly understood, since technology alone does not solve problems. If people do not know how to adapt to change, how to use the technology, which part of the process they are involved in or how to use the process well, technology will not provide the best return on investment and BC may be compromised [97,98].

Table 5 contains subtopics that have a more holistic view of the organization, but are related to BC. According to the British Standards Institution (BSI), "Organizational resilience is the ability of an organization to anticipate, prepare for, respond and adapt to incremental change and sudden disruptions in order to survive and prosper" [99]. This helps us to understand that the balance in the people/processes/technology grouping helps create resilience-driven solutions which, in turn, help to enforce GRC in organizations, thus making these topics highly relevant [8,100]. This would, for example, help to avoid disruption in organizations by preparing rapid response/recovery protocols [101]. Understanding which digital or technological threats and impacts are most relevant for better governance of an organization is also on the agenda in the literature [16].

**Table 4.** Group of key findings related to people/processes/technology.

| | Key Areas/Subjects for Investigation | Count * |
|---|---|---|
| **People** | Capabilities to support and ensure BC | 69 |
| | Knowledge work/acquisition/management | 16 |
| | Human-in-the-loop | 10 |
| **Processes** | Change management | 23 |
| | Framework (conceptual/risk/control) | 23 |
| | Guidelines | 15 |
| | Legislation | 8 |
| | Workflows (composition and system) | 8 |
| | Standards (create/implement) | 7 |
| | Building responding processes and guidelines | 7 |
| | Tools/technology/skills | 4 |
| | Security orchestration research and practice | 2 |
| **Technology** | Emerging technologies (types/impacts) | 53 |
| | Regulatory | 8 |
| | Regulatory framework | 3 |

* Number of articles found related to PPT and their subtopics.

**Table 5.** Key findings for BC.

| | Key Areas/Subjects for Investigation | Count * |
|---|---|---|
| **Business Continuity** | Resilience-driven solutions | 64 |
| | Governance of new digital/technological threats and impacts | 47 |
| | Disruption avoidance | 33 |
| | Preparing just-in-case scenarios | 8 |
| | Rapid response/recovery protocols | 7 |

* Number of articles related to business continuity and their subtopics.

## 7.1. Research Areas to Investigate

The authors are aware that this investigation has limitations which future research will need to address. Hence, in line with the research approach, a set of areas to investigate in the future was compiled, resulting from an analysis of the outcomes found in the results and future work sections of the literature. Figure 7 was structured and presented according to three central elements of our research themes (GRC/PPT/BC). These topics are organized in relation to RPA/IPA and BC and have been arranged according to their meta-theme/cluster (GRC/PPT). They emerge from the themes for future investigation indicated in the articles that were analyzed or themes that raised concerns and need more development, compiled from the results found in the articles. In presenting the future areas for investigation, we have added examples from various contexts to make these areas tangible.

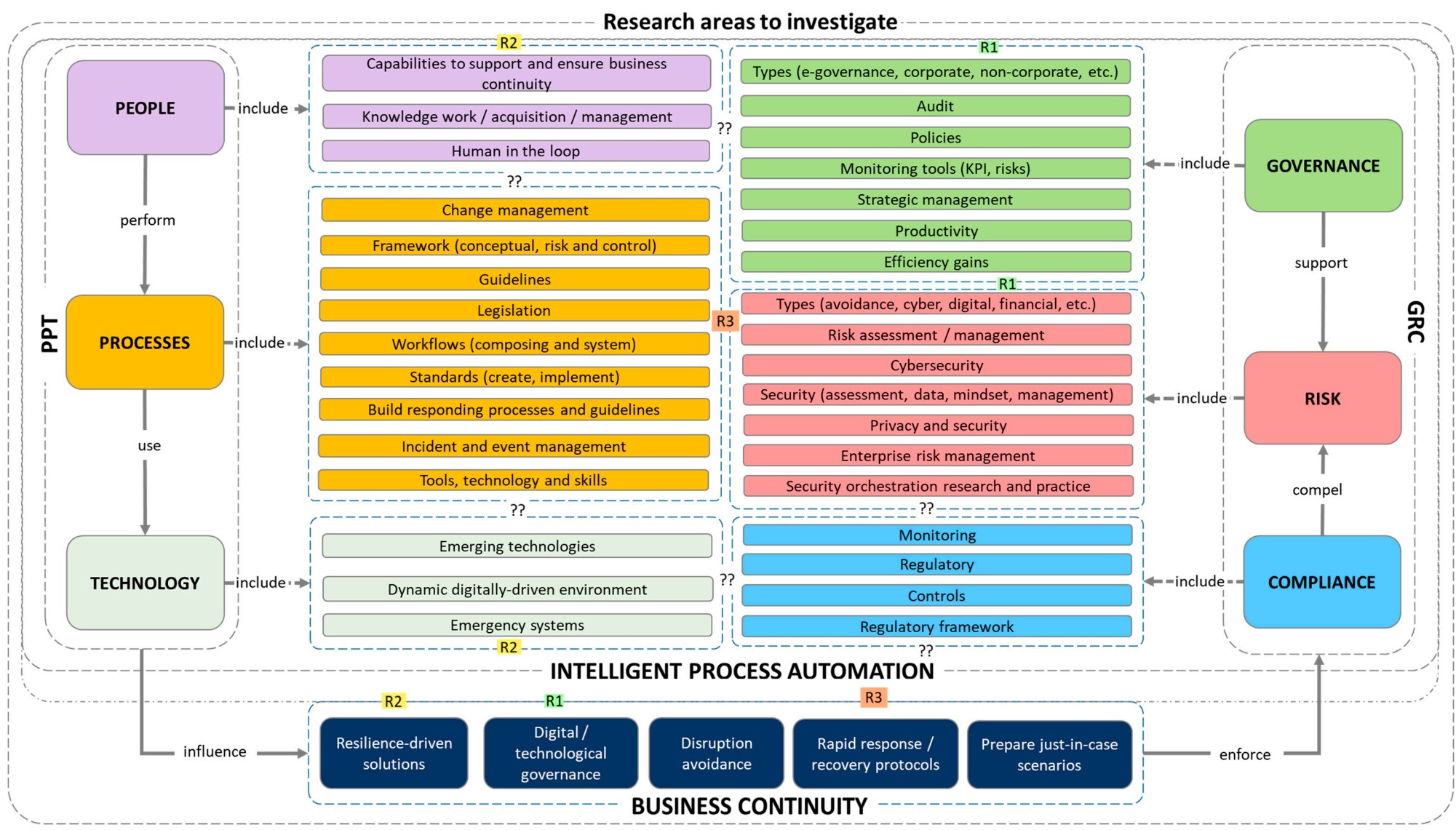

**Figure 7.** Main areas identified for further investigation.

### 7.1.1. Governance/Risk/Compliance

In the past, governing boards and directors, as well as senior management, could delegate, ignore or avoid ICT-related decisions. Nowadays, due to the fact that DT and ICT have become crucial to the support, sustainability and growth of enterprises, this topic has become a priority that can no longer be ignored [16,24,102–113].

#### Types of Governance

A total of 27 papers on this topic show that organizations often underestimate the challenges associated with integrating RPA into their operations, which can leave them vulnerable to risks and facing additional challenges in implementing controls, therefore resulting in governance problems [67,70,101,114]. The solution for mitigating risk in RPA is to follow a strict governance program, and audit rules and controls must therefore be defined correctly [24,92,94,115–124].

#### Audit Plans

The implementation of process automation programs using RPA/IPA leads to exposure to high risks compared to typical IT automation processes. By analyzing some cases from an audit perspective, we uncovered that there are clear changes in process risk definitions after automation changes in different job functions, impacting access security, considerations related to application change, strategy, and governance of the RPA environment. With more complex environments, auditing/assurance of these ET is becoming more complex than a regular technology audit [26,89,92,118,120,125–135].

#### Internal Policies

The incorporation of RPA/IPA makes it necessary to adapt or create new internal policies to be integrated into the BC policy [9,50,136–140]. Organizations should document all policies and processes and make this information centrally available so that it can be used for self-learning and training [1,92,132,141].

#### Types of Risks

A total of 71 results report that some new and disruptive technologies are not well known or understood and can present a range of unknown types of risks if no mitigation response is prepared in advance [142–147]. On the one hand, some literature points to an increased risk arising from the adoption and use of new technologies and the lack of knowledge of their real impact on organizations, while other authors point to evidence that technology helps to mitigate many types of risk [137,148–159]. In addition, organizations need to comply with internal and external regulations and therefore need mechanisms that enable them to respond to legislative concerns about their process activities. This allows them to develop a policy approach focus that reflects technological change, evolving to respond to legislative frameworks and regulatory standards to mitigate risks that arise from the adoption of new technologies or any other situations [110,160,161]. The Digital Operational Resilience Act (DORA) is a concrete realization of these concerns applied to financial institutions [162,163], pinpointing the need to find and implement automated control mechanisms that are fast and flexible to audit, report, share information, and monitor organizations so that their data and information workflows can be understood. It enables their impact on the organization's risk profile to be controlled, by enforcing the implementation of security measures and related testing controls when dealing with information and communication technology (ICT), which also includes RPA/IPA [161,164,165].

#### Risk Assessment/Management

While RPA can reduce unintended or intentional human errors, adopting RPA/IPA introduces new risks that companies need to understand and address [96,134,166–169]. One of the many risks analyzed comprises those related to cybersecurity, the failure of the organization to consider the effects of operational changes on its internal controls,

or forgetting to update its BC plans [64,65,110]. Failure to properly assess/identify and manage these new risks can erode or limit the entire value created by the adoption of RPA/IPA [170]. In order to grasp the full impacts of automation, companies must consider how RPA affects risk across multiple categories and, at the same time, use its potential to detect those same risks [70,91,171]. For example, AI is attracting great interest as companies explore its ability to unlock value through improved revenues, customer service, efficiency and risk management [112,114,120,132,172].

It is imperative that companies re-evaluate existing BCPs, conduct thorough risk assessments, and identify new vulnerabilities imposed by ET and recent changes in the way we work [29,52,90,94,153,158,172–176].

Monitoring Solutions

An RPA can monitor an extended set of systems in real time according to a wide range of policies and controls [13,140]. BC with continuous monitoring mechanisms is necessary to guarantee the quality associated with the execution of processes, namely for the monitoring of models associated with cognitive components and, at the same time, the integration of AI components in RPA processes, from state control of robots to monitoring changes in their performance [13,48,70,71,94,166,177–179].

As RPA is subject to errors, and resources are needed to ensure detailed monitoring of their operations. This cannot be achieved by using manually monitored controls or a 'wait to see' approach. Monitoring and exception handling mechanisms are needed, and business process management systems (BPMSs) can be used to support these types of issues. In addition to the data collected by RPA/IPA, BPMSs can gather additional information on the execution, duration of execution or properties associated with compliance. This facilitates monitoring by revealing when an RPA process does not behave as expected and by creating alerts leading to immediate maintenance actions on detection of bugs or changes in the requirements of the source applications [87,89,99,124,132,133,179–190].

7.1.2. People/Processes/Technology

Based on the results obtained, there is a perception that organizations are showing a growing interest in seeking out what is new and what is to come in technology [111,177,191,192], although they need to be able to balance these three elements (PPT) to maintain a good relationship between them [97]. This implies preparing the workforce (people) to perform their activities (processes) and choosing the correct tools (technology) to be included [9,24,84,90,104,109,119,120,129,136,141,160,170,178,193–217].

Emerging Technologies

This topic is highlighted in our research in terms of technology, for example in the relevance of AI that is indicated for leveraging IA by combining the strengths of RPA/IPA, AI and human intelligence [63,65,68,105,161,169,218–225]. Now that organizations are beginning to implement technology that has gone beyond the proof-of-concept phases into live systems, the demand for a structured framework to ensure competitiveness, while also guaranteeing the ability to meet the demands of security, regulatory compliance, change management, rapid response to disruptive events and integration with current systems, has become the order of the day [3,10,42,65,90,91,96,129,141,170,178,179,195–200,226–246].

Change Management

References were found signaling the importance for organizations to prepare human workers, not only to provide a solid framework for cooperation with RPA/IPA, but also to mould mentalities and attitudes in relation to change management and new learning [9,119,212,247,248].

In implementing IA, it will be necessary to provide a basis for designing the future workforce that will manage this new reality and also to prepare employees for

the consequences of automation by formulating an appropriate response to managing change [69,201].

An suitable roadmap with all the steps needed to implement the changes required to incorporate new technology will avoid risks associated with IT systems, lack of knowledge, operations and BC (since RPA/IPA do fail and crash) [94,120,126,212,249].

Guidelines

With regard to BC, the Business Continuity Institute (BCI) has compiled a document containing the current national and international legislation, regulations and standards for business continuity management [207]. To ensure resilience, organizations need to have a vision of their capabilities, threats and impacts associated with RPA/IPA and business continuity, and to develop a knowledge structure that will provide guidelines for each business unit that uses automation in the execution of its processes [12,85,140,201,228,250].

Capacity to Support and Ensure BC

Furthermore, 69 of the publications argue that it will be necessary to train teams to support the new processes and activities with new capabilities to support ET and, at the same time, ensure BC [1,3,25,36,45,62,88,92,107,118,136,169,174,184,193,197,198,202–204,211,222,231,233–237,251–293].

The interaction between humans and the automation aspects of BP is also a concern, and guidelines are necessary to regulate these new activities and avoid negative impacts due to neglect or misuse of the technology. Automation projects are not just limited to changing processes and technology, since this necessarily entails fundamental changes in terms of human resources, involving updated skills and responsibilities to respond to new challenges [42,48,97,131,136,148,175,202,231,294–296]. The initial incorporation of these initiatives, together with a properly organized change management program, is of paramount importance to addressing and embracing the human side of the digital modernization initiative. This is critical to avoid disruption and ensure the business continuity [3,47,48,62,69,72,92,97,107,118,131,168,177,277,297–299].

7.1.3. Business Continuity

COVID-19 highlighted the fact that most organizations did not have adequate measures to prepare for BC and disaster recovery [24,34,44,268,300,301], and organizations now realize that they need to find and implement solutions to make them more resilient [172,290,302–305]. RPA platforms, for example, can help and are recommended for making operations more intelligent and for building IT and business resilience [214,215,270,304,306–309].

According to the literature that was reviewed, DT can be an enabler of enterprise resilience since, if it is implemented correctly, it will increase connectivity, transparency, collaboration and innovation [26,261,310–315].

Resilience-Driven Solutions

In order to become resilient, an organization needs to have the ability to anticipate problems and know how to prepare to respond or adapt to sudden changes or interruptions so that it will survive or thrive in the event of disruptive situations [13,25,34,43,47–49,52,82,90,92,94,99,113,213,226,227,229,247,316–321].

Automation adapts to changing market dynamics among organizations, employees and their customers in real time. Operations support processes are also being revamped and will be adapted so that they can respond more quickly to constantly changing environments [88,152,175,182,193,197,222,230,235,253,322–326].

Digital resilience will rely on several areas, including the following:

- The transformation of manual financial services processes (banking/insurance), to respond to changes in market dynamics with greater flexibility, effective communication and confidence;

- Optimizing resource management using automation and predictive analytics to determine where technicians and maintenance personnel should be deployed;
- Enabling rapid business DT with adaptable and resilient business models to enhance relationships with customers, and ensure they benefit from greater value in the process outsourcing services provided, through deeper collaboration and co-innovation [51,85,100,194,262,268,269,279,280,295,306,307,310,314,326–336].

Governance of New Digital/Technological Threats and Their Impacts

A DT project involves multiple factors in the transformation of the business model, impacting the entire organization—especially operational processes, resources and internal and external users—since it involves a major change in habits and ways of working, based on collaboration and intensive interactions [140,213,259,260,313,326,331,337–346].

These changes present different types of impacts as well as threats, such as those linked to the incorporation of ET smart products and services, as well as the ways in which organizations interact with their customers, improve operational efficiency, increase revenue, strengthen the competitiveness of offerings and improve customer experience. In order for organizations to have a clear vision of what to expect regarding these changes, it will be necessary to reassess the risks and their governance practices [93,96,142,153,156,180,198,232,254,304,347–349].

A properly structured program is required for identification and protection against digital risks, as part of a unified DT plan that includes several elements, such as the selection of appropriate digital technology and its implementation, and protection measures against digital threats from new technologies: moreover, it must clearly understand their (positive/negative) impacts [4,9,14–16,46,47,52,53,90,91,99,121,129,136,177,350,351].

Disruption Avoidance

Avoiding disruption to a particular business service support system or process can be seen as a contributing factor to operational resilience. However, the phenomena that lead to disruptions are very complex and non-linear and no satisfactory model has yet been developed to avoid them or predict when they will occur and what kind of impacts they will cause. For this reason, the ML and AI techniques associated with RPA/IPA have begun to be widely used in recent years [8,10,26,85,104,115,131,145,148,152,176,201,204,215,217, 260,272,278,290,295,299,304,311,312,314,325,329–331,352–357].

*7.2. Findings*

Figure 7 illustrates the proposed framework and is composed of three meta-themes or clusters: governance, risk management, and compliance (GRC); people, processes and technology (PPT); and business continuity (BC). These clusters suggest further investigation in order to develop an even more detailed structural research approach to RPA/IPA in combination with business continuity. The framework was developed to incorporate future research areas. It should be noted that the inter-domain touchpoints, shown as "??", represent the existing uncertainties to explore, and R1, R2 and R3 are relations that have already been discussed in the literature but need to be evaluated and further investigated.

All the topics that are synthesized in Figure 7 are findings that were organized according to the topics of governance, risk, and compliance (GRC), and to people, processes, and technology (PPT). All of them are associated with both robotic process automation and business continuity, showing the direct interest evidenced in the literature with how to provide resilient solutions to allow businesses to continue to operate in case of disruption.

The PPT cluster in Figure 7 shows the group of people that performs processes that include technology, emphasizing that organizations need to incorporate in-the-loop knowledge workers with the right capabilities/functionalities to support and ensure BC. Moreover, these workers will need to use processes based on updated guidelines, legislation, standards and applicable frameworks. The technology enables ET such as RPA/IPA, AI or blockchain and will balance the two other elements (people and processes) by lever-

aging their capabilities while, at the same time, creating solutions for BC that allow for organizational resilience.

Another priority emerges from the literature on the enforcement of GRC in relation to BC topics, which points to preparing organizations to be more resilient and responsive. Although there are only a few specific topics regarding BC, all the reviewed literature refers to the association between RPA/IPA/Automation and BC; in this way therefore, all the constituents of the macro topics contribute in the same way to BC.

The analysis of the GRC cluster revealed quite a few references to topics of interest, such as the regulatory framework, control, regulatory and monitoring solutions that define organizational compliance when applying policies, relevant laws, and regulations. Risk is the macro topic that is attracting most interest in the literature found, and hence it is necessary to assess which types of risk are raising the most concerns—privacy, security avoidance, cyber or financial—in order to detect which threats could have a negative impact on an organization's ability to conduct business.

Finally, the results for governance in terms of number of publications indicate that in order to steer an organization towards BC, topics such as auditing, policies, monitoring tools (applied to GRC), strategic management, productivity and efficiency gains, are on the agenda.

Although it was not the main goal of this research, some evidence of relationships could be found in the literature. They include governance/cyber risks and business continuity, expressed as R1 [93], which highlights the need for qualifications for all levels of the business in emerging areas of risk in the preparation of business continuity plans, and governance for cyber-attacks, both in the short and long term as a preventive measure. Another relation found, R2, concerns the role of emerging and intelligent technologies in the design and development of responsive supply chains, enhancing their capacity to ensure business continuity [197]. R3 is related to the publication of a legislative proposal to create a digital operational resilience framework for the EU financial services sector. It relates legislation, guidelines, and incident and event management from the PPT cluster, on the one hand, to the GRC cluster for risk assessment, cyber and digital risk themes present in the business continuity cluster [162], on the other.

## 8. Conclusions

This study presents an MLR on RPA/IPA correlated with BC, aiming to identify the state of awareness of RPA/IPA within the BCM lifecycle and to highlight areas for future research. In the course of our research, we realized that digital solutions providers have started to publicize and present RPA and IA as a business continuity enabler [220,320,321,358–360].

For professionals, this survey identifies themes that should be on the agenda for BC and DT using ET. RPA/IPA can be very useful and can leverage daily operations but may contain risks that need to be properly understood, monitored, controlled and addressed. Hence, organizations should update their policies and guidelines to ensure they reflect audit capabilities that allow for secure monitoring of all the changes that are being incorporated into their businesses. Another important aspect concerns the need to provide the necessary tools, knowledge and skills to their members, so that they are able to face new challenges that may arise due to the incorporation of certain kinds of ET.

In providing an initial conceptualization of the interplay between RPA/IPA and business continuity, as well as presenting and discussing related areas for future research, we hope our results stimulate broad discussion within the community on the possible adaptations of processes that have received little attention to date, but which we consider highly relevant.

The highlighted results are not based on the total number of articles found per cluster, as many of the articles are associated with a variety of themes, but on the theme that stands out most in each case.

In the case of the GRC cluster, the subtopic with the most results was the risk topic— "Types of risks" to be analyzed, associated with IPA/RPA and business continuity. In the PPT cluster, it was the capabilities associated with people and RPA/IPA required to

support BC that was most underlined. Finally, in the BC cluster, a large number of articles highlighted the need to find resilience-oriented solutions. These themes will be prioritized in our future research.

Figure 7 summarizes the findings and can guide academics towards topics that merit further investigation, namely:

- Which threats and impacts organizations may expect when incorporating RPA/IPA into businesses;
- Mapping risks emerging from the use of RPA/IPA and finding ways to mitigate them;
- Resilience-driven solutions for dealing with known threats and impacts, in particular those emerging from the use of RPA/IPA;
- The new capabilities associated with RPA/IPA that are more useful in terms of supporting and ensuring BC;
- Ways to incorporate governance measures to deal with new RPA/IPA threats and impacts and avoid disruption;
- Correlation of all the identified topics with BC and RPA or IPA, in order to find solutions that can mitigate possible risks, while at the same time taking advantage of their benefits;
- ICT support for all the innovative and complex systems used in this new digital age. Greater digitization, automation, interconnectivity, and also their interdependency, amplify ICT risks, making society as a whole—and the financial system in particular—more vulnerable to cyber threats or ICT disruptions. Although the universal use of ICT systems and the high level of digitization, automation and connectivity (where RPA/IPA is included) are currently essential features of all global activities, digital resilience is still not sufficiently integrated into the operational frameworks of organizations, and there is therefore a need for research into how to evolve in this area [106,162].

As with many other studies, ours has limitations which future research needs to address. Most significantly, the areas presented for future investigation were derived from the individual contributions of researchers working in RPA/IPA and BC and from grey literature. While we cannot formally claim the integrity and validity of our results, our approach is in line with common standards and guidelines for conducting qualitative research. Nevertheless, future research should explore these areas of investigation more rigorously (e.g., as the subject of Masters' theses, using exploratory interviews, focus groups, or the Delphi method).

**Author Contributions:** Conceptualization, J.B. and R.P.; methodology, J.B.; validation, R.P. and S.M.; formal analysis, J.B.; investigation, J.B.; resources J.B.; writing—original draft preparation, J.B.; writing—review and editing, J.B. and Ruben; visualization J.B.; supervision, R.P. and S.M.; project administration, R.P. All authors have read and agreed to the published version of the manuscript.

**Funding:** This research received no external funding.

**Data Availability Statement:** No new data was created.

**Conflicts of Interest:** The authors declare no conflict of interest.

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
