# Peer review of "Intelligent Process Automation and Business Continuity: Areas for Future Research"

_information, doi:10.3390/info14020122_

Round 1
Reviewer 1 Report
This work presents a literature review on process automation, correlated with business continuity. To begin at the beginning, preliminary concepts are clustered and hierarchically organized, also considering inter-domain touchpoints. Many pictures are used to explain correlations and results. For the purposes of this study, authors prefer to use a multivocal literature review approach, for taking into consideration also what they call "grey" literature like preprints, white papers, and the like. Results of the review are presented throughout all key findings. Finally, the manuscript ends with a discussion on research areas for investigation, with a big clear picture on this issue.
I find the study rather interesting both for professionals and academics. There are indeed many points of interest: the literature review research is well conducted – as far as I know, I'm not an expert, the state of the art seems inclusive, and I did not find any other work to mention –, inclusion and exclusion criteria seem reasonable, the narrative is pleasant and mature, the results are well organized and presented, and – this, I really find a strength – the overall pictures are more than simply descriptive, they are really efficient in triggering the reader interest and understanding.
For all these reasons, I strongly suggest to accept this work in its present form.
Author Response
Dear reviewer,
Many thanks for your feedback and kind words.
Grateful for the attention given.
Reviewer 2 Report
Dear authors,
Happy to read your article.
Please address following points to improve your article.
Figure 2 doesn’t illustrate upon “9/11 in the USA, the Islamic state (ISIS) 74 terrorist attacks throughout the world, climate changes that affect the planet, and the recent COVID 19 pandemic” Please provide references.
Line 131: “Both scientific research and the media highlight the potential of RPA for increasing the efficiency of processes.” Please provide reference.
Line 137: “The first approach to this technology (Process Automation)” Do you mean Intelligent Process Automation?
Figure 3: seems RPA and IPA are being used interchangeably, please clarify in section 2.2.
Figure 4 contradicts with lines 180-184.
Figure 7 needs to be explained more clearly in section 6.2 that is the crux of this article.
Regards,
Author Response
Dear Reviewer,
Thank you very much for your feedback with all designated points for improvement. In this way, we were able to improve the work presented.
We send you a new version in the hope of meeting your expectations, as well as the aspects pointed out by the editor.
Grateful for the attention given.

Reviewer 3 Report
The topic is interesting and the methodology of the literature review is appropriate. My only concern is that all important research questions are left for future investigation as mentioned in the last section of the paper. Given the length of the paper, I would like to leave my concern to the Editor to decide whether a revision is necessary.
Author Response
Dear reviewer,
Thank you very much for your feedback with all designated points for improvement. In this way, we were able to improve the work presented.
We send you a new version in the hope of meeting your expectations, as well as the aspects pointed out by the editor.
Grateful for the attention given.
